# European Corn Borer (*Ostrinia nubilalis* Hbn.) Bioecology in Eastern Romania

**DOI:** 10.3390/insects14090738

**Published:** 2023-08-31

**Authors:** Paula Lucelia Pintilie, Elena Trotuș, Nela Tălmaciu, Liviu Mihai Irimia, Monica Herea, Ionela Mocanu, Roxana Georgiana Amarghioalei, Lorena Diana Popa, Mihai Tălmaciu

**Affiliations:** 1Agricultural Research and Development Station Secuieni–Neamț, Principala St, 371, Secuieni, 617415 Neamt, Romania; paula.ursache@scda.ro (P.L.P.); roxana.amarghioalei@scda.ro (R.G.A.); dy.hemp420@gmail.com (L.D.P.); 2Department of Plant Protection, Faculty of Horticulture, University of Life Sciences, Mihail Sadoveanu Ally, no 3, 700490 Iași, Romania; livirimia@uaiasi.ro (L.M.I.); mherea@uaiasi.ro (M.H.); imocanu@uaiasi.ro (I.M.); mtalmaciu@uaiasi.ro (M.T.)

**Keywords:** climatic change, population, dynamics

## Abstract

**Simple Summary:**

The European corn borer, *Ostrinia nubilalis* Hbn., is an annual pest of the Romanian corn crop. The research followed the development of the European corn borer in the context of the climate warming recorded in recent years in Eastern Romania. We monitored the development of the egg, larva, pupa, and moth stages using cage studies and observations in an experimental cornfield. We also analyzed the impact of climatic conditions on developmental time. Our data show that the increase in winter temperatures ensures the survival of a higher percentage of mature larvae, which leads to intense moth flight at the end of June and the beginning of July. However, the high summer temperatures caused an increasing percentage of the larvae to transform into pupae, which led to the recording of a secondary flight in August. Although the insect is known to have one generation per year, it is apparent that a multivoltine ecotype is also present. We conclude that the climate warming we are witnessing affects the species’ development and that higher larval infestations could occur.

**Abstract:**

Between 2020 and 2021, we conducted research in eastern Romania to monitor the bioecology of the European corn borer (*Ostrinia nubilalis* Hbn.), an important pest of corn. The bioecology research established the pest stage duration (egg, larva, pupa, and moth), the flight curve, and the flight peak. The bioecological study occurred in the experimental corn field and a field cage. According to our findings, the insect has one generation per year. The European corn borer hibernates as a mature larva in corn residues and continues developing in the spring, when the weather warms. It pupates from May to July over 37 days. Analyzing the data recorded during the winters of 2020 and 2021, we observed that the warming trend favored the high survival of hibernating larvae (60.7%). Due to the large number of mature larvae that had favorable conditions during the winter, there is an intense flight, starting in June and ending in September. When the first moth was caught in the light trap, the ∑(tn-10) °C (sum of degree days) was 245.6 °C. In 2020, the flight was recorded for 94 days. In 2021, the European corn borer flight lasted 104 days. The initial egg masses were detected when the total of ∑(tn-10) °C reached 351.5 °C. Moths laid the eggs for 25 days, mostly during peak flight in late June and early July. The first larvae hatched when ∑(tn-10) °C totaled 438.4 °C, and stages III–V were recorded in the harvested crop. Understanding the bioecology of the European corn borer can offer valuable insights into managing population levels and identifying optimal timing for addressing infestations in corn crops.

## 1. Introduction

Corn (*Zea mays* L.) is one of the most important crops worldwide and equally important in Europe, where the temperate climate offers favorable conditions for growing corn over large areas. Thus, the corn crop occupies 28.25 million ha in the northern half of the continent, 2.75 million ha in the southern part of the continent, and 2.41 million ha in the western regions, while in the east, it occupies 14.0 million ha [1]. Average yields vary widely between 5449 and 6829 kg/ha in the eastern and northern regions of Europe and between 8556 and 9296 kg/ha in the southern and western areas of the continent [1].

As with each agricultural crop, corn is affected by numerous pests [2]. One of these with an essential impact on corn yield is the European corn borer (*Ostrinia nubilalis* Hbn., Lepidoptera: Crambidae) [3,4], a pest where the larval feeding injury disrupts nutrient flow within the plant, causing significant yield loss [5,6]. In Romania, the pest is present in all areas where corn is the main crop [7]. The larvae feed on the epidermis of the leaves and the pollen from the inflorescences, and as it passes through the larval stages, it pierces the stem, the veins of the leaves, and the cobs, consuming the grain within the cob [8,9]. Larval boring causes the plants to break, which makes mechanized harvesting difficult, leading to direct quantitative losses and indirectly to grain deterioration. The attack favors the introduction of fungi of the genus *Fusarium*, which depreciates the quality of the grains [10].

The European corn borer’s global presence shows its high adaptability to different climate zones in Europe and America [11]. In Europe, the number of complete generations for the European corn borer differs depending on the region; in the Mediterranean countries, the insect has two–three generations. In the central part of the continent, one–two generations occur, and in the northern regions, one generation per year [12,13]. Research carried out in Romania [14] showed that, depending on the average annual temperature, the European corn borer develops one generation per year in the northern part of the country. A partial second generation occurs in the country’s south [8,9]. Recent research indicates that rising temperatures have created favorable conditions for this species to develop in the country’s south for two complete generations per year [15]. In countries neighboring Romania, the European corn borer has two generations in Serbia [16,17] and one or, sporadically, two generations in Poland [18].

Areas where the European corn borer has two or more generations suffer high economic losses due to the aggressiveness of the second generation, which creates galleries in the stem and cobs [19,20]. The appearance of a second complete generation is closely related to the increase in growing days and the extension of the vegetative period of the host plants. According to Trnka et al. [21], the increase of +0.2 to +0.9 °C in average annual temperatures since the 1990s led to numerous high-intensity attacks in the Czech Republic. According to the authors’ estimates, with the increase in temperature [21], by 2025, the flight will begin 4 to 10 days earlier, and it will shorten the European corn borer biological cycle by 9 to 15 days. Another study [22] shows that the areas where the univoltine population exists are becoming more significant. At the same time, the possibility of the appearance of a second generation has increased by 61%, leading to more numerous and intense attacks. The population increase is due to the interaction between climatic factors (long, warm, and dry autumns, followed by mild winters without a layer of snow and low temperatures) and agronomic factors (ignoring cultural hygiene measures such as chopping of vegetable remains and monoculture) [23]. The increase in mean annual temperature affects insect bioecology by extending its spatial distribution to the north, increasing the number of generations, reducing the diapause time, and extending the development period of insects [24].

For several years, a monitoring program for the European corn borer was implemented in Romania using different pheromone traps. This research found that insect appearance is affected by spring climate conditions, and tracking insect flight dynamics revealed that the attack could impact the entire corn crop [25]. More specifically, the frequency of attack varies depending on the areas: in the eastern regions, it produces an average frequency of attack of 30.3% [9]; in the center and western regions, the frequency of attack is close to 59% [26,27]; in the southwest regions, it is between 22.4% and 40.7% [28], while in the south of the country, the frequency of attack varies between 50% and 70% [29].

This work aimed to analyze the influence of recent climatic conditions on the bioecology of the European corn borer in the eastern part of Romania. The results could help establish the most appropriate measures to manage the pest in corn.

## 2. Materials and Methods

### 2.1. Experimental Site

We carried out the research within the Agricultural and Development Research Station Secuieni–Neamț, located at 26°5′ east longitude and 46°5′ north latitude, in a hilly area where the altitudes go up to 250 m above sea level (a.s.l.). The region’s topography includes extensive interfluvial plains, meadows, and terraces [30]. The area has a temperate continental climate, Dfb in the Koppen-Geiger climate updated classification, with short springs, cool summers, and harsh winters [31].

The research developed between 2020 and 2021, when an experimental cornfield of 2.0 ha was established by sowing the Turda Star, a semi–early hybrid with a vegetation period of 110–115 days (maturity group FAO 370), which was obtained at the Turda Agriculture Research Station in Romania [32]. The genotype is characterized by a good tolerance to the lower temperatures that register in the first phases of the plant’s development. The plant is tall and has 13–15 semi-erect leaves. The plants tolerate hot periods, pest attacks, and diseases well. The predecessor plant was the sunflower. The autumn plowing was carried out at 20 cm depth, and in the spring, passes were made with the disc harrow and the harrow with adjustable tines. The germinal bed was prepared before sowing, with the soil being well–leveled and shredded. Fertilization was carried out with 200 kg a.s./ha of complex fertilizer (20:20:0). Sowing was effectuated at the end of April with the Kleine seeder at a distance of 70 cm between the rows and 22 cm between seeds for an optimal seeding density of 68,000 g.g./ha; plant harvesting was carried out in October. No insecticide treatments were applied to the vegetation.

### 2.2. Bioecology Methods

The bioecology research established the duration of the pest’s stages (egg, larva, pupa, and moths), the flight curve, and the maximum moth flight peak. Observations and determinations were carried out using a field cage [11]. The field cage size was 2.5 m × 2.5 m × 1.5 m, covered with netting that prevented other insects from entering. In the spring, corn was sown inside the cage to monitor each developmental stage’s appearance and duration, registering the appearance of the pupae, moths, egg-laying, larvae, and larval attack mode (Figure 1).

The light trap was used to monitor the insect’s flight and consisted of a collector tube, collector vessel, and lighting source (250-watt neon) placed in a wooden cage, preventing rainfall degradation [18,33]. The light source was turned on overnight, and chloroform moistened the tissue. The trap was checked in the morning when the insects were collected in Petri dishes, labeled with the day’s date, determined by species in the laboratory, and registered in the light trap register.

### 2.3. Bioecology Determinations

The egg stage is recorded on corn plants in the field under natural conditions while capturing the first adults, and moth flights are monitored using a light trap. For egg identification, determinations were performed by examining 25 consecutive plants in five randomly chosen places in the corn field (250 plants) on each of the two days. Sampling was carried out diagonally, starting from 10 m inside the field. The organs of each plant were inspected for egg mass: two leaves above the cob, two leaves below the cob, and the corn cob. The plants where the egg mass was identified were marked with a white thread to monitor the eggs’ development. The dates of the first and last eggs laid were registered to establish the period in which the eggs were laid. The egg mass development was analyzed at each determination and classified as usual stage/viable, dehydrated/dry, or hatched.

The larval stage was further monitored in field conditions on the plants presenting the insect’s egg mass, registering the appearance of the first, the last, and the attack mode. Sections were made on plants that showed signs of an attack to establish the stage duration and register the larval development. For larva identification, this study was completed by field observation, where ten randomly infested plants were selected from the field in three replicates and sectioned out every week from early July until the crop harvest.

To establish the diapause period, at the end of the corn vegetative period, 150 stalks with larvae were collected, placed in the cage, and left there until spring. The observations continued in the spring to determine the survival percentage of the larvae over the winter, the transformation into pupae, and the stage duration (recording the first and last pupae transformations).

The pupal stage was determined by analyzing the larvae from the 150 stalks stored in the cage. The stems that had larvae inside were checked for the appearance of the first and last pupae and their development, establishing the stage duration.

The determinations established the moth stage based on the captures collected daily from the light trap. The start and end of the flight, as well as the maximum flight peak, have been established.

### 2.4. Larval Attack Parameters at Corn Crops

Simultaneously with monitoring the insect bioecology, the larvae’ attack on corn crops was determined. At the end of the vegetative period, before harvesting, samples of 150 attacked plants were collected and sectioned in three repetitions to assess the frequency of attacked plants, the average number of holes, galleries, and larvae on the plant, and the average length of galleries (cm). Larval attack parameters are established as follows:-the frequency of attacked plants (%) is obtained by reporting the number of attacked plants to the total number of plants;-the average number of holes/plant is obtained by counting the holes on the surface of the plants and relating them to the number of attacked plants;-the average number of galleries/plant is, in fact, the number of galleries inside the plants, which were counted and related to the number of attacked plants;-the average number of larvae/plant is represented by the number of larvae inside the galleries, which were counted and related to the number of attacked plants;-the average length of the galleries (cm) is a comparison between the galleries that were measured and related to the number of identified galleries.

### 2.5. Statistical Analysis

The results obtained in 2020 and 2021 are presented as average values of the monitored parameters: frequency of attacked plants, average number of holes/plant, average number of galleries/plant, average number of larvae/plant, and average length of galleries (cm). At the end of the corn’s vegetative period, we assess the frequency of attacked plants in the cornfield (50 plants in ten repetitions). To evaluate the remaining parameters, 150 attacked plants were assessed in three repetitions. The experimental data obtained were analyzed by appropriate statistical methods using the least significant differences (LSD) test (*p* < 0.001—***/OOO—positive very significant/negative very significant; *p* < 0.01—**/OO—positive distinct significant/negative distinct significant; *p* < 0.05—*/O—positive significant/negative significant).

### 2.6. Climate Conditions

Climate data were recorded using a Wireless Vantage Pro 2 Plus weather station (SC Rom Tech SRL, Sibiu, Romania) located in the experimental field. The data recorded by the weather station are the maximum daily temperature (°C), the minimum daily temperature (°C), the average daily temperature (°C), the daily rainfall (mm), the relative humidity (%), the maximum daily soil temperature (°C), the minimum daily soil temperature (°C), and the average daily soil temperature (°C). To characterize the years from a climatic point of view, we used the data related to the average air temperature (°C) registered at 2 m high and the rainfall ∑ (mm).

To correlate the climatic conditions with the European corn borer bioecology, we calculated the sum of effective temperatures ∑(tn-10 °C), and we followed the temperatures recorded from January 1st of each year until the first pupa appeared, the first adults were captured, the first eggs were identified, and the first larvae hatched. For each stage, the differences between the average temperature of each day and 10 °C were computed and summed up [34,35]. Moreover, the maximum and average temperature °C in the air and the average temperature °C at ground level were calculated separately. The thermal and pluviometrical conditions during the development stages were characterized based on data recorded by the weather station.

## 3. Results

### 3.1. Climatological Analysis of the Area for the 2020–2021 Time Period

Temperature recordings for the 2020–2021 time period revealed a warming trend, with average annual temperature deviations of +2.3 °C in 2019–2020 and +1.0 °C in 2020–2021, compared to the multiannual average of 8.9 °C (Figure 2). Moreover, during the 2019–2020 period, 9 out of 12 months recorded higher average temperatures; the deviations ranged from +1.2 °C for June to 5.6 °C for February. The monthly deviations of the average temperatures for 2020–2021 ranged from +3.6 °C for October to −2.0 °C for April.

Regarding the pluviometry, the region’s climate registered a deficit, with the annual rainfall ∑ (mm) being reduced by −168.3 mm in 2019–2020 and by −145.5 mm in 2020–2021, compared to the multiannual average of 544.3 mm (Figure 2). For the 2019–2020 period, the driest months were April (45.7 mm), July (43.3 mm), and December (−19.2 mm), while for the 2020–2021 period, the most deficient months in rainfall were November (7.4 mm), January (12.2 mm), February (10.8 mm), and September (9.2 mm) (Figure 2).

The climatic conditions for the 2020–2021 time period influenced not only the development of the corn crop but also the appearance, spread, and attack of the European corn borer. From a thermal point of view, the increase in the average temperature of the summer months (June, July, and August) of 2020 and 2021, compared to the multi-year average, is noticeable. They were warmer, with average monthly temperatures higher by +1.2 °C and +2.7 °C, respectively, as compared with the multiannual values, except for July 2020 and June and August 2021, when deviations did not exceed 1 °C as compared to the multiannual average. The rainfall was reduced, with the three summer months being less dry in 2020, while in 2021, June was normal, July was very dry, and August was less rainy.

All these climatic conditions made the summer of 2020 less favorable and the summer of 2021 very favorable for the European corn borer’s development and attack.

### 3.2. European Corn Borer Bioecology

#### 3.2.1. Pupae

The mature larva developed into the pupal stage inside the stem; the first pupae were identified in late May (2020) or early June (2021). The last larva that turned into a pupa was identified at the beginning of July. At the emergence of the pupae stage, the ∑(tn-10) °C was 127.1 °C in 2020 and 161.8 °C in 2021 (Table 1). The larval development into pupae occurred gradually over 37 days in 2020 and 38 days in 2021. The average temperature during stage development was 18.7 °C in air and 21.7 °C in soil in 2020 and 22 °C in air and 23.5 °C in soil in 2021 (Table 1).

#### 3.2.2. Moth Stage

The moth’s flight was different each year.

In 2020, the first moths of the European corn borer were captured on June 10, with the flight continuing until the end of September for 94 days (Table 1). The maximum flight peak was recorded at the end of June (97 moths/trap) and mid–August (106 moths/trap) (Figure 3). During the European corn borer flight, the average air temperature was 19.1 °C, and the maximum was 25.8 °C (Table 1). At the moment of moth appearance, on June 10, the ∑(tn-10) °C was 228.5 °C, while the RH for June–August was 71–82% (Figure 4).

In 2021, the European corn borer moths began to fly in mid–June, when the first moths were captured. The ∑(tn-10) °C was 262.6 °C (Table 1). Relatively low average temperatures in May and the first part of June led to the extension of the pupal stage and created unfavorable conditions for the moth’s emergence. The flight lasted 104 days; during this interval, the average air temperature was 19.7 °C, the maximum temperature was 26.6 °C, and the RH was between May and August, 79–94%. The flight peak was recorded in the middle of July (182 moths) and mid–August (Figure 5). 

#### 3.2.3. Egg

Observations in the field showed that most eggs were laid on leaves above and below the cob and in smaller numbers on the remaining leaves. Oviposition under field conditions occurred during the maximum flight peak; the first was identified on June 21 and June 25 (Table 1). Egg masses were identified in the experimental field for 24–26 days. The egg mass appeared at ∑(tn-10) °C 342.9 °C in 2020 and 360.1 °C in 2021 (Table 1).

#### 3.2.4. Larvae

In autumn, the mature larvae enter hibernation inside the gallery they create. Mild climatic conditions during the winter favor the survival rate of mature larvae (Table 2). The diapause duration varied between 207 days in 2020 and 212 days in 2021. On average, of the total number of mature larvae identified in the spring, the percentage of alive larvae was 60.7%, while 40.3% were dead. The soil temperature varied during the two years between 6.4 ℃ and 6.8 ℃, while the average air temperature was 4.5 ℃ between November 2020 and June 2021 and 5.7 ℃ between November 2019 and May 2020.

#### 3.2.5. Corn’s Larval Attack

In 2020, the frequency of attacked plants was 42.54%, and the larvae produced, on average, 0.81 holes/plant. After sectioning the plants, it was found that the average number of galleries was 0.33 galleries/plant, which shows us that most of the galleries joined and the larvae created several entry or exit points on the plant (Table 3).

Larvae created galleries of 7.29 cm long, and 0.61 larvae/plant were recorded in those galleries at the end of the vegetative period (Table 3).

In the following year, the attack rate was much higher, the frequency of the attacked plants was 82.10%, and it can be seen that the rest of the parameters registered higher values compared to the previous year. The number of holes/plant was 2.27, indicating that the attacked plants recorded at least two holes on the plant surface (Table 3).

Regarding the average number of galleries created by the larvae, it can be seen that their number increased; an average of 1.72 galleries/plant was recorded, and the number of larvae identified was higher, with at least one larva being found in the galleries of the attacked plants. The larvae created much longer galleries than the previous year, on average 33.76 cm (Table 3).

In the environmental conditions of Eastern Romania, the European corn borer passes during the 2020 and 2021 years through all stages of development: egg, larva, pupa, and adult (Figure 6). From the first pupae’s appearance to the larva’s withdrawal for hibernation, the ∑(tn-10) °C summed up to 1342 °C (Figure 6).

## 4. Discussion

We found that the temperature increases expedited or prolonged certain stages in the European corn borer’s development. One of the changes was the appearance of the second peak in August. The number of generations in Eastern Romania was stable for 25 years at one generation [9]; however, in the last few years, a partial second generation registered because of the temperature increases. However, these changes in insect development take place over time, with rising temperatures reported during the last few years and an increase in effective heat units, or degree–days, accumulating per day and annually above the biological thresholds [36]. The results suggest that the insect adapts to regional conditions, and the number of generations varies according to the environmental gradients interacting with the populations [37]. Moreover, there is a difference between the moth’s flight and the number of generations since not every flight peak means a new generation [38].

In Romania’s southern and western regions, insect flights are monitored due to their intense attacks, which cause issues for corn growers [7,15]. In this region, two generations are recorded, the first in June and the second at the beginning of August [7]. In the central area of Transylvania, insect monitoring is carried out annually, with the insects being present from June until September [39]. In the last few years (2018–2020), due to consistent rains during the growing season, the attack frequency was between 23 and 33% [40,41]. In the conditions of Eastern Romania, in the last 25 years, the average attack of European corn borer was between 23.8% and 27.8% [8].

The European corn borer successfully develops all stages and presents one generation per year and, in some years, due to climatic conditions, a partial second generation. It found a close connection between the temperatures recorded annually and insect flight. The research carried out by Trotus et al. in Moldova [8] shows that intense flight was recorded in warm years (48%), followed by normal years (38%), while the cool years were the most unfavorable for moths flying (19%). It was observed that in the Eastern part of the country, the highest percentage of moths were captured during normal years (49%), followed by dry (27%) and rainy (24%) years, with annual rainfall being analyzed [8]. In the 2018 and 2019 years, the moth flight was continuous, with the warming trend in the area recording an annual trend of 1.1 to 1.4 °C above the multiannual average of 8.9 °C, while the yearly sum rainfall registered deviations of 18.0 mm (2018) and −114.1 mm (2019) as compared to the multiannual average of 544.3 mm [42].

When the conditions are favorable, some of the larvae pupate, and a second flight appears. Still, the eggs laid by the females do not have favorable conditions of temperature and humidity for the larvae to hatch. In their study, Frolov and Grushevaya [43,44] found that some of the larvae, up to 30%, pupated, and a new generation appeared. The same conclusion was reached by Barbulescu [45,46,47], who observed that in southern Romania’s conditions, 20% of the larvae transform into pupae, and a second generation of moths appears. Other researchers also present this situation when the second generation appears, claiming that the larvae turned into pupae without entering diapause. The case is favored by the high temperatures during the summer [48].

In retrospect, if the development of the first generation is not affected by drought stress, the second generation will face a deficit of soil and atmospheric water, leading to changes in egg hatching. This is due to relative humidity being reduced to 50% and the mating process, which will reduce the hibernating larvae population [21]. A spring with rainfall influences insect development, increasing moth prolificacy and creating the conditions for the second generation that emerges from the first eggs [49]. In our study, the first moth’s appearance was different: in 2020, the moths appeared at the beginning of June, while in 2021, the low temperatures and the small rainfall sum recorded in May and at the beginning of June led to a delay, with the flight starting at the second part of June (Figure 3, Figure 4 and Figure 5).

Our research shows that mild winters characterized by low rainfall and increased temperatures of 3.3 to 5.6 °C (2020) and 1.8 to 3.4 °C (2021) favor the survival of the hibernating larvae over the winter. According to Chiang [50], 18% of the mature larvae overwinter in the cobs, and 82% overwinter in the stems left in the field. Therefore, plant residues with mature larvae represent the most important source that ensures the continuity of the population. The transformation into pupa of mature larvae can reach 90% in warm years, compared to cool years when the percentage of transformation decreases [36]. During our research, the years were warm and dry, leading to a higher rate of larvae surviving the winter, so we recorded a more intense European corn borer flight.

Winter temperatures have an impact on diapause length. This period ensures that the larva’s metabolism slows down. High temperatures shorten the diapause and lead to increased energy consumption by the larvae, which causes the premature end of the diapause.

Popović et al. [51] and Kojić et al. [52] support through their published work the importance of low temperatures during diapause when different biochemical transformations take place in the larva’s body and show that high temperatures during diapause lead to premature death of larvae, which means higher mortality of larval populations.

Comparing our results with Popović et al. [51] and Kojić et al. [52], we found that high temperatures shortened the diapause period in our case, and the first pupae appeared earlier. Uzelac et al. [53] showed that during diapause, the activity of metabolic enzymes is closely related to temperatures close to 0 °C. At the time of oviposition by females, a vital factor in their development is the relative humidity of the air and the maximum temperature during the day. For European corn borer, the maximum oviposition took place at the end of June or beginning of July. This period corresponded to the peak flight and, in conjunction with the phenology of the crop plant, when the corn was in the ear stage, beginning to flower and silken. In our study, the appearance of most larvae coincided with pre-tasselling, tasselling, and silk, followed by plant blooming, phenophases in which the newborn larvae feed on plant pollen. Kuhar et al. [54] found that wasps of the genus *Trichogramma* parasitize some eggs of the corn borer and persist in the corn crop throughout the growing season when pest hosts are abundant.

When reviewing our data, the years were warm and dry, leading to a higher rate of larvae surviving the winter, so we recorded a more intense European corn borer flight. In 2021, it started later than 2020 due to a cool spring, which delayed the female’s maximum flight peak and egg laying. Oviposition continued until the middle of July. However, the larval appearance was staggered, and the attack produced by it was much more visible. By contrast, the results of Beres [55] in Poland showed that the shortest oviposition period of females occurred in hot and dry years, with rainy and cold years extending the oviposition period. In Croatia, Lemic’ et al. (2019) [56] recorded the first moths in early May, not in June as expected. Sarajlic et al. (2017) [57] observed that years with high summer temperatures and a lack of rainfall favor rapid insect development and lead to a higher attack. In rainy and cool years, insect development is delayed by up to 15 days, and the damage is significantly reduced. The Kornoşor and Kayapinar study [58] on the development of European corn borer in the Kukorova region in Turkey shows that the development time of each stage varies according to climatic conditions (temperature and humidity).

The climatic conditions represented by humidity, temperature, and rainfall contributed to the shortness or prolongation of the mature larval transformation into pupae, the period of moth emergence, and larval hatching, respectively. Water is an important source for moths; rainy springs increase the chances of mating by increasing the lifespan of moths and female fertility [59], while Kira et al. (1969) [60] point out that larvae are sensitive to dehydration in the absence of water.

Our study has limitations since it only refers to certain areas, and environmental conditions and the population of the European corn borer in these areas influence the development of the species. Therefore, the presented results were influenced by multiple factors that somewhat limit the applicability of the data.

One is the limited research period; however, the studies must be extended longer. Long-term studies are necessary to accurately determine if the borer population in our area is multivoltine. We can observe a mix of univoltine and multivoltine, highlighted by the appearance of an extra flight of European corn borer, influenced by the increase in summer temperatures.

Another factor that can affect the validity of the data is the food source. Climate change has a significant impact on species’ development that rely primarily on maize as a host plant. The effects are evident both in accelerating the vegetative period and reaching maturity.

The European corn borer larvae can turn to other hosts, such as hemp and sorghum. Still, these crops are also negatively impacted by rising temperatures and do not provide ideal conditions for development.

Our method relied on visually inspecting the plants to determine the insect stages and identify any signs of attacks. Our data on the stages of occurrence reflect the field situation and specific pedoclimatic conditions at that time. It is important to consider the number of plants analyzed when determining the stages of occurrence and development. The plants attacked at the end of the vegetation period reveal a detailed picture of the larvae’s impact. The larvae create holes in the plants to penetrate inside, create galleries to feed, and leave behind a trail of damage. Some larvae survive until the end of the vegetation period, while others perish, seek out softer tissues to feed on or turn into pupae.

It is important to take into account the insects’ ability to adapt to the climate when implementing pest control measures in corn crops.

## 5. Conclusions

In Eastern Romania, the European corn borer bioecology closely depends on climate conditions, especially temperature and rainfall. Our study indicates that the European corn borer has two generations per year: a complete one between June and September and a partial one in August. Our research also confirmed that the source of the second moth flight was attributed to the portion of the larval population in late summer that pupated in preparation for moth eclosion versus the part that typically remains in the field as diapausing larvae in preparation for overwintering. Some larvae undergo direct transformation into pupae, resulting in a partial generation. This is because the average temperature is 1.0 to 2.3 °C higher than the 25-year multiannual average of 8.9 °C. The research found that insect appearance is affected by spring climate conditions, and tracking insect flight dynamics revealed that the attack could impact the entire corn crop.

## Figures and Tables

**Figure 1 insects-14-00738-f001:**
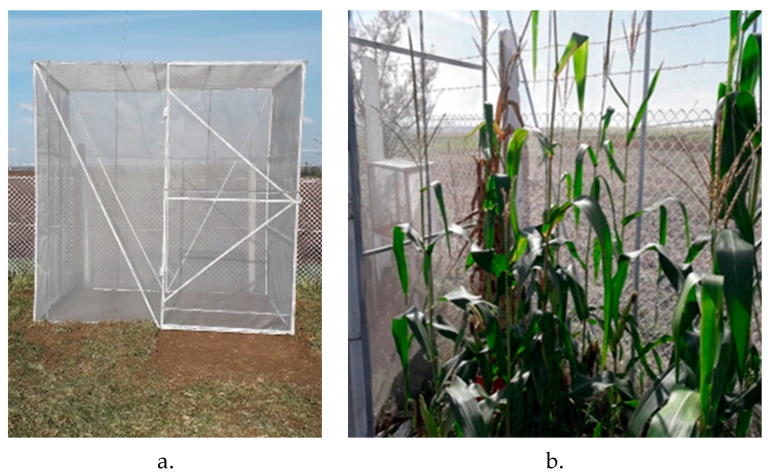
Walk–in field cage: outside (**a**) and inside view (**b**).

**Figure 2 insects-14-00738-f002:**
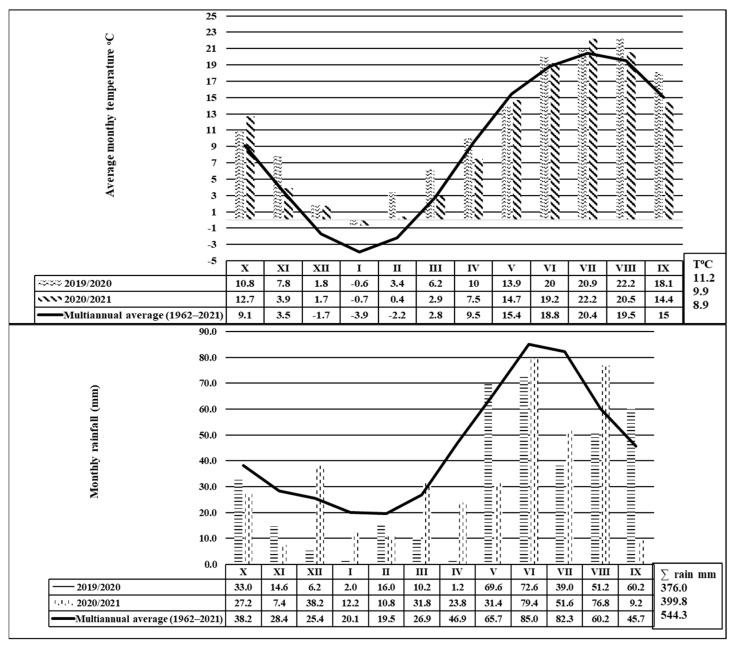
Monthly temperatures (°C) and rainfall (mm) in 2019–2020 and 2020–2021.

**Figure 3 insects-14-00738-f003:**
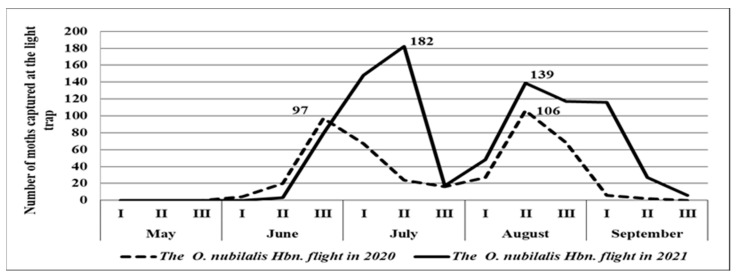
European corn borer flights in 2020 and 2021.

**Figure 4 insects-14-00738-f004:**
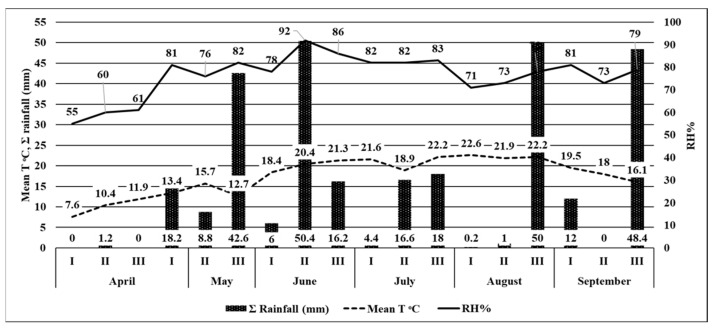
The climatic conditions for the April to September 2020 time period. (RH (%): relative humidity values; Mean T °C: mean temperature values; ∑ rainfall (mm): rainfall values)

**Figure 5 insects-14-00738-f005:**
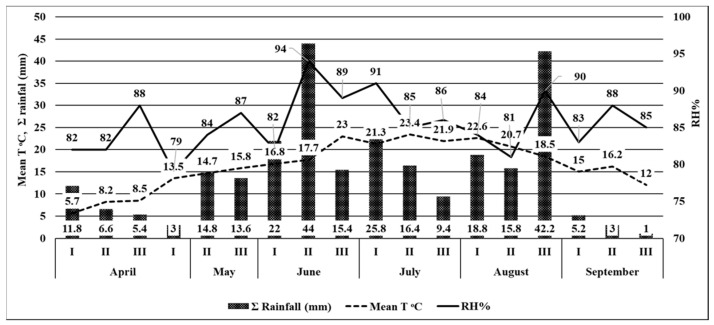
The climatic conditions for the April to September 2021 time period. (RH (%): relative humidity values; Mean T °C: mean temperature values; ∑ rainfall (mm): rainfall values)

**Figure 6 insects-14-00738-f006:**
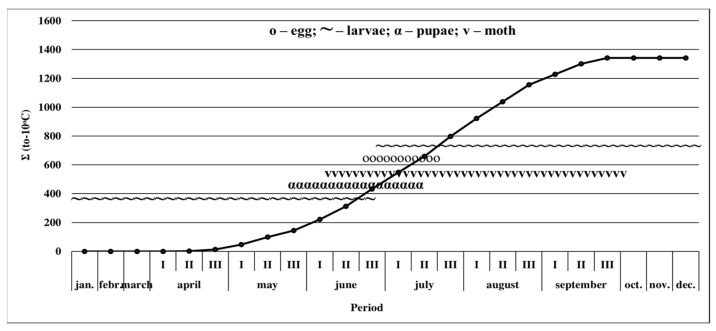
European corn borer biological cycle in the conditions of Eastern Romania (average 2020–2021). (Note: the solid line = ∑(tn-10) °C is the sum of degree °C above 10 °C calculated from January 1st until December 31; January–December = monitored period.

**Table 1 insects-14-00738-t001:** European corn borer stages in the conditions of Eastern Romania (averages 2020–2021).

Year	Stage	The Period When the Stage Was Identified	Stage Period (Days)	∑(tn-10) °C * at Stage Initiation	Average Temperature during Stage Development			∑ pp (mm)	RH%
					Max. Air °C	Avr. Air °C	Soil °C		
2020	Pupae	25.05–30.06	37	127.1	24.8	18.7	21.7	111.8	87
	Moth	10.06–12.09	94	228.5	25.8	19.1	-	168.6	82
	Egg	21.06–14.07	24	342.9	27.7	21	-	30.4	83
	Larvae	29.06–04.11	129	434.7	25	17.8	-	179.8	82
2021	Pupae	04.06–12.07	38	161.8	27	22	23.5	92.4	91
	Moth	17.06–29.09	104	262.6	26.6	19.7	-	17.3	87
	Egg	25.06–20.07	26	360.1	29.3	22.5	-	55.6	88
	Larvae	02.07–06.11	127	442	23.3	16.3	-	150.2	86
Average 2020–2021	Pupae	May–beginning of July	37.5	144.5	25.9	20.4	22.6	102.1	89
	Moth	June–September	99	245.6	26.2	19.4	-	93.0	84.5
	Egg	End of June–Mid–July	25	351.5	28.5	21.8	-	43	85.5
	Larvae	June (2020)–July (2021)	128	438.4	24.2	17.1	-	165.0	84

* ∑(tn-10 °C) is the sum of differences between the average temperature of each day and 10 °C, calculated from January 1st of each year until the first stage of identification; the average and maximum air, respectively, and soil temperature values are the temperatures recorded at the weather station of the unit during the interval in which the development of each stage took place.

**Table 2 insects-14-00738-t002:** European corn borer larvae mature in Eastern Romania (averages 2020–2021).

Year	The Interval during Which the Mature Larval Stage Was Identified	Stage Period (in Days)	Plant Parts Analyzed	% Living Larvae	% Dead Larvae	Average T °C Soil	Average T °C Air
2020	November 2019–May 2020	207	100	54.3	45.7	6.8	5.7
2021	November 2020–June 2021	212	100	67.1	34.8	6.4	4.5
Average	2020–2021	210	100	60.7	40.3	6.6	5.1

**Table 3 insects-14-00738-t003:** The larvae attack corn crops.

No	Year	Larvae Attack				
		Frequency of Attack (%)	Average Number of Holes/Plant	Average Number of Galleries/Plant	Average Number of Larvae/Plant	Average Gallery Length (cm)
1	2020	42.54 ^ooo^	0.81	0.33	0.61	7.29
2	2021	82.10 ***	2.27	1.72	1.93	33.76
3	Average (Ct)	62.32	1.54	1.03	1.27	20.53
LSD 5%		4.19				
LSD 1%		5.96				
LSD 0.1%		8.63				

***/^ooo^—positive very significant/negative very significant.

## Data Availability

The data presented in this study are available on request from the corresponding author.

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
