# Peer review of "European Corn Borer (Ostrinia nubilalis Hbn.) Bioecology in Eastern Romania"

_insects, 2023, doi:10.3390/insects14090738_

Round 1

Reviewer 1 Report

Specific comments:

LINE 142  Add manufacturers and models of the light trap. Black or incandescent lamps?

Unfortunately, the limitations, novelty and contribution of the work is still not enough. Here are some published papers that are still not available for comparative analysis and discussion:

Kuhar, T.P.; Wright, M.G.; Hoffmann, M.P.; Chenus, S.A. Life Table Studies of European Corn Borer (Lepidoptera: Crambidae) with and without Inoculative Releases of Trichogramma Ostriniae (Hymenoptera: Trichogrammatidae). Environ. Entomol. 2002, 31, 482–489, doi:10.1603/0046-225X-31.3.482.

Hudon M, LeRoux EJ. 1986a. Biology and population dynamics of the European corn borer (Ostrinia nubilalis) with special reference to sweet corn in Quebec. I. Systematics, morphology, geographical distribution, host range, economic importance. Phytoprotection 67: 39-54.

Hudon M, LeRoux EJ. 1986b. Biology and population dynamics of the European corn borer (Ostrinia nubilalis) with special reference to sweet corn in Quebec. II. Bionomics. Phytoprotection 67: 81-92.

Hudon M, LeRoux EJ. 1986c. Biology and population dynamics of the European corn borer (Ostrinia nubilalis) with special reference to sweet corn in Quebec. III. Population dynamics and spatial distribution. Phytoprotection 67: 93-115.

Kornoşor, S.; Kayapinar, A. Biology and life tables of European corn borer (Ostrinia nubilalis Hbn., Lep.: Pyralidae) causing damage on corn in Cukurova Region. Turk. J. Entomol. 1988, 12.

Author Response

Hello,

Thank you for all your suggestion. We  have replied to your comments and please see the attachment.

Reviewer 2 Report

Comments and Suggestions for Authors

Manuscript ID: insects-2475897

The paper entitled “Research regarding the European corn borer (Ostrinia nubilalis Hbn.) bioecology in Eastern Romania” was carefully reviewed. This paper aims to monitor the development of O. nubilalis and to provide new insights into the bioecology of this insect in eastern Romania.

It is important to note that the field study was conducted in a specific region (Eastern Romania) and may not be generalizable to other regions or countries. Additionally, the research was conducted over a specific time period and may not reflect the current conditions or trends. This study was conducted using specific methods and may not account for all possible factors that could impact the bioecology of O. nubilalis. These limitations should be taken into consideration in the present paper when interpreting the findings and applying them to other contexts.

The manuscript would really benefit from proofreading by an English editing service or a native English speaker as some sentences are very difficult to understand and there are many errors and typos.

Detailed comments:

Title

-          The title is too vague and inappropriate, when it should be concise and informative.

Introduction

-          Line 61: Start the paragraph by: “Ostrinia nubilalis Hbn.” instead of “O. nubilalis Hbn.”

-          Line 73: Correct “Areas were O. nubilalis Hbn. has…” by “Areas where O. nubilalis Hbn. has…”.

-          Line 77: Replace “the increase of +0.2 …+0.9 °C in…” by “the increase of +0.2 to 0.9 °C in…”.

-          Line 80: Replace “Other research…” by “Another research…”.

-          Line 93: “More specific”, by “More specifically,”.

Materials and methods

-          Replace entomological isolator by cage throughout the text, since we're talking about field experiments.

-          Line 204: delete the following sentence as it is not relevant: “The data for each experiment were processed in Excel (Microsoft USA)”.

-          The authors should improve the description of the statistical analysis section and provide more information on the data distribution and the tests used for each parameter. It is important to check the normality of the data before applying parametric tests such as ANOVA and LSD/student test.

-          The authors state that “The obtained results were processed with the ANOVA (analysis of variance – Fisher test) and the limit differences – LSD (student test where …)”. I’m confused here. Did you use LSD or Student test?

-          The authors should also explain the meaning of the symbols (***/OOO; **/OO; */O) and how they indicate the significance levels. Alternatively, the authors could use standard notation such as p < 0.05, p < 0.01, and p < 0.001 to report the results of the statistical tests.

-          Line 216: Correct the following sentence: “For the O. nubilalis Hbn. bioecology was calculate the…”.

-          Line 220: How were the maximum and average temperature in the air and the average temperature at ground level calculated?

Results:

-          Figure 4 compares the climatic data for two time periods. Correct the title of this figure as follows: “Monthly temperatures and rainfall in 2019-2020 and 2020-2021”.

-          Line 258: Reorganize the results subsection (3.2. O. nubilalis Hbn. Bioecology) as follows: Eggs, larvae, pupae, moths.

Discussion:

-          Lines 352-353: Delete the following sentence to avoid redundancy: “This study aimed to obtain data and findings relevant for the Eastern part of Romania, regarding the influence of climate on the O. nubilalis bioecology”.

-          Many studies have been conducted and published worldwide on the bioecology of O. nubilalis. The following papers should be considered to enrich the discussion section:

·         Uzelac, I., Avramov, M., Čelić, T. et al. Effect of Cold Acclimation on Selected Metabolic Enzymes During Diapause in The European Corn Borer Ostrinia nubilalis (Hbn.). Sci Rep 10, 9085 (2020). https://doi.org/10.1038/s41598-020-65926-w

·         Popović ŽD, Maier V, Avramov M, Uzelac I, Gošić-Dondo S, Blagojević D, Koštál V. Acclimations to Cold and Warm Conditions Differently Affect the Energy Metabolism of Diapausing Larvae of the European Corn Borer Ostrinia nubilalis (Hbn.). Front Physiol. 2021 Nov 22;12:768593. doi: 10.3389/fphys.2021.768593.

·         Danijela Kojić, Željko D. Popović, Dejan Orčić, Jelena Purać, Snežana Orčić, Elvira L. Vukašinović, Tatjana V. Nikolić, Duško P. Blagojević, The influence of low temperature and diapause phase on sugar and polyol content in the European corn borer Ostrinia nubilalis (Hbn.), Journal of Insect Physiology, Volume 109, 2018, Pages 107-113, https://doi.org/10.1016/j.jinsphys.2018.07.007

·         Bažok, R.; Pejić, I.; Čačija, M.; Virić Gašparić, H.; Lemić, D.; Drmić, Z.; Kadoić Balaško, M. Weather Conditions and Maturity Group Impacts on the Infestation of First Generation European Corn Borers in Maize Hybrids in Croatia. Plants 2020, 9, 1387, doi:10.3390/plants9101387.

·         Hudon M, LeRoux EJ. 1986a. Biology and population dynamics of the European corn borer (Ostrinia nubilalis) with special reference to sweet corn in Quebec. I. Systematics, morphology, geographical distribution, host range, economic importance. Phytoprotection 67: 39-54.

·         Hudon M, LeRoux EJ. 1986b. Biology and population dynamics of the European corn borer (Ostrinia nubilalis) with special reference to sweet corn in Quebec. II. Bionomics. Phytoprotection 67: 81-92.

·         Hudon M, LeRoux EJ. 1986c. Biology and population dynamics of the European corn borer (Ostrinia nubilalis) with special reference to sweet corn in Quebec. III. Population dynamics and spatial distribution. Phytoprotection 67: 93-115.

·         Kuhar, T.P.; Wright, M.G.; Hoffmann, M.P.; Chenus, S.A. Life Table Studies of European Corn Borer (Lepidoptera: Crambidae) with and without Inoculative Releases of Trichogramma Ostriniae (Hymenoptera: Trichogrammatidae). Environ. Entomol. 2002, 31, 482–489, doi:10.1603/0046-225X-31.3.482.

·         Kornoşor, S.; Kayapinar, A. Biology and life tables of European corn borer (Ostrinia nubilalis Hbn., Lep.: Pyralidae) causing damage on corn in Cukurova Region. Turk. J. Entomol. 1988, 12.

The manuscript would really benefit from proofreading by an English editing service or a native English speaker as some sentences are very difficult to understand and there are many errors and typos.

Author Response

Hello, 

Thank you for all your suggestion. We have replied to your comments, please see the attachment.

Reviewer 3 Report

Authors provide a detailed analysis of the ecology of ECB in Romania, and documentation of a "multi-voltine" ecotype in this region. There may be a mix of Univoltine and Multivoltine in the region, but this is difficult to determine. The Multi- is evident with the "extra flight of ECB" at the end of season, when warmer summer/autumn weather occurs, and ECB larvae do not stay in diapause but proceed to pupate, produce moths. 

Overall, flow outline of results is acceptable, but English use must be improved. Detailed edits are included in the attached file ---and only for the pages that require edits. Thus the need for major revision. Many edits are provided to assist the authors.

References Cited:  

Numerous edits were made to correct formatting for Ref Cited section (see encl. pdf). In addition, the following new, 2023, paper should be added, and cited in the Introduction (and Disc. sections); (has EU focus) i.e.,

Kaçar, G., A.M. Butrón Gomez,  D. Kontogiannatos, P. Han; M. Fernanda, G.V. Peñaflor; Gema P. Farinós, Fangneng Huang, et al. 2023. Recent trends in management strategies for two major maize borers: Ostrinia nubilalis (Lepidoptera: Crambidae) and Sesamia nonagrioides (Lepidoptera: Noctuidae). J. Pest Science. (Review). 96: 879–901. https://link.springer.com/article/10.1007/s10340-023-01595-8

Many copy edits are necessary to improve the English, writing style. I do not have time to make all edits. Authors should seek an English speaking colleague to assist with revising the manuscript.

Author Response

Hello,

Thank you for your comments and suggestion, We replied to them, please see the attachment.

Round 2

Reviewer 2 Report

Comments and Suggestions for Authors

Manuscript ID: insects-2475897

The revised version of the paper entitled “European corn borer (Ostrinia nubilalis Hbn.) bioecology in Eastern Romania” was carefully reviewed. This paper aims to monitor the development of O. nubilalis and to provide new insights into the bioecology of this insect in eastern Romania.

Detailed comments:

Materials and methods

-          This comment has not been taken into account in the revised version of the manuscript: “Replace entomological isolator by cage throughout the text, since we're talking about field experiments” (see lines 26, 167 & 172).

-          This comment has not been taken into account in the revised version of the manuscript: The authors should improve the description of the statistical analysis section and provide more information on the data distribution and the tests used for each parameter. It is important to check the normality of the data before applying parametric tests such as ANOVA and LSD/student test (see lines 198-203).

-          This comment has not been taken into account in the revised version of the manuscript: “The authors should also explain the meaning of the symbols (***/OOO; **/OO; */O) and how they indicate the significance levels. Alternatively, the authors could use standard notation such as p < 0.05, p < 0.01, and p < 0.001 to report the results of the statistical tests” (see line 202).

Results:

-          This comment has not been taken into account in the revised version of the manuscript: “Reorganize the results subsection (3.2. O. nubilalis Hbn. Bioecology) as follows: Eggs, larvae, pupae, moths” (see lines 256 to 349).

Discussion:

-          I strongly recommend that authors include a separate section (2-3 paragraphs) at the end of the discussion to explicitly state the limitations of the study and how they may affect the validity and generalizability of the results. For example, authors could mention potential sources of error or bias in the methods, such as sample size, design or measurement. They could also discuss how other factors they have not taken into account, such as soil quality, crop variety or pest management practices, might impact on the bioecology of O. nubilalis.

Conclusion:

-          Lines 456-458: “In the research conducted in the Eastern part of Romania, the bioecology is closely related to the climatic conditions during the year, mainly the temperature level and rain- 457 fall”. The bioecology of O. nubilalis? Please correct the sentence as you have only studied one species.

-          Make your conclusion more effective and impactful. Avoid repeating the same information or using the same words multiple times, such as “the increased temperatures and rainfall deficit during the summer months lead to the registration of a complete generation, to which a partial one adds” and “the insect had a complete generation and a partial one.

Quality of English Language

-          Once again, the manuscript would benefit from proofreading by an English editing service, as some sentences (grammar, spelling and punctuation) are still very difficult to understand and there are many errors and typos.

Once again, the manuscript would benefit from proofreading by an English editing service, as some sentences (grammar, spelling and punctuation) are still very difficult to understand and there are many errors and typos.

Author Response

REVIEW REPORT 2 ROUND 2

Comments and Suggestions for Authors

Manuscript ID: insects-2475897

The revised version of the paper entitled “European corn borer (Ostrinia nubilalis Hbn.) bioecology in Eastern Romania” was carefully reviewed. This paper aims to monitor the development of O. nubilalis and to provide new insights into the bioecology of this insect in eastern Romania.

 Thank you for your time and your valuable comments and suggestions for authors.

Detailed comments:

Materials and methods

-          This comment has not been taken into account in the revised version of the manuscript: “Replace entomological isolator by cage throughout the text, since we're talking about field experiments” (see lines 26, 167 & 172).

Done it. LINES 14, 26,129,131,137,165,170,

-          This comment has not been taken into account in the revised version of the manuscript: The authors should improve the description of the statistical analysis section and provide more information on the data distribution and the tests used for each parameter. It is important to check the normality of the data before applying parametric tests such as ANOVA and LSD/student test (see lines 198-203).

I have added additions to this section. Lines 195-204

-          This comment has not been taken into account in the revised version of the manuscript: “The authors should also explain the meaning of the symbols (***/OOO; **/OO; */O) and how they indicate the significance levels. Alternatively, the authors could use standard notation such as p < 0.05, p < 0.01, and p < 0.001 to report the results of the statistical tests” (see line 202).

The meaning of the symbols is listed at lines 202-204 and under the table 3 (line 335)

Results:

-          This comment has not been taken into account in the revised version of the manuscript: “Reorganize the results subsection (3.2. O. nubilalis Hbn. Bioecology) as follows: Eggs, larvae, pupae, moths” (see lines 256 to 349).

Thank you for the advice. It is more concise.

Discussion:

-          I strongly recommend that authors include a separate section (2-3 paragraphs) at the end of the discussion to explicitly state the limitations of the study and how they may affect the validity and generalizability of the results. For example, authors could mention potential sources of error or bias in the methods, such as sample size, design or measurement. They could also discuss how other factors they have not taken into account, such as soil quality, crop variety or pest management practices, might impact on the bioecology of O. nubilalis.

Thank you for the advice. Lines 449-474

Conclusion:

-          Lines 456-458: “In the research conducted in the Eastern part of Romania, the bioecology is closely related to the climatic conditions during the year, mainly the temperature level and rain- 457 fall”. The bioecology of O. nubilalis? Please correct the sentence as you have only studied one species.

-          Make your conclusion more effective and impactful. Avoid repeating the same information or using the same words multiple times, such as “the increased temperatures and rainfall deficit during the summer months lead to the registration of a complete generation, to which a partial one adds” and “the insect had a complete generation and a partial one.

I rewrote the conclusions taking into account your suggestion above. Lines 477-487

“In Eastern Romania, the European corn borer bioecology closely depends on climate conditions, especially temperature and rainfall. Our study indicates that the European corn borer has two generations per year, a complete one between June and September and a partial one in August. Our research also confirmed that the source of the 2nd moth flight was attributed to the portion of the larval population in late summer that pupated in preparation for moth eclosion versus the part that typically remains in the field as diapausing larvae in preparation for overwintering. Some larvae undergo direct transformation into pupae, resulting in a partial generation. This is due to the average temperature being 1.0 to 2.3°C higher than the 25-year multiannual average of 8.9°C. The research found that insect appearance is affected by spring climate conditions, and tracking insect flight dynamics revealed that the attack could impact the entire corn crop.”

.Quality of English Language

-          Once again, the manuscript would benefit from proofreading by an English editing service, as some sentences (grammar, spelling and punctuation) are still very difficult to understand and there are many errors and typos.

Comments on the Quality of English Language

Once again, the manuscript would benefit from proofreading by an English editing service, as some sentences (grammar, spelling and punctuation) are still very difficult to understand and there are many errors and typos.

Thank you for your suggestions. The manuscript has been revised again by an English editing service and contains other suggestions and additions made by reviewer 3, so some sentences or paragraphs have been rewritten to be more concise.

Reviewer 3 Report

The authors did a nice job with the first round of revisions; the ms contains very useful data for how ECB populations are changing with regard to voltinism, and documentation of high, damaging infestations. My remaining necessary edits, include:

a) Several final English useage, grammar and/or terminology edits have been made directly to the ms, via Word tracking feature; these edits must be made before the paper can be published.

b) The authors need to be very careful regarding their use of generations, voltinism ecotypes; given their data, what they are seeing is a Mix of Both a Univoltine strain (that still appears to be the dominant strain), combined with a Multivoltine strain, that is known to have a facultative diapause (see the Palmer et al. 1985 ref.); this MV strain is most noticeable late season (Aug-Sept), and at this latitude of Romania, may only produce a partial moth flight, but it is consistent each year. (Much more could be said about this; but is beyond the scope of this study)

Key point: is be careful with the terminology; and be consistent throughout the paper

With the revisions made to the attached Word file, the paper should be acceptable for publication.  

Much better; but see additional edits

Author Response

REVIEW REPORT 3 ROUND 2

Comments and Suggestions for Authors

The authors did a nice job with the first round of revisions; the ms contains very useful data for how ECB populations are changing with regard to voltinism, and documentation of high, damaging infestations.

Thank you for your valuable comments and suggestions.

My remaining necessary edits, include:

  1. a) Several final English useage, grammar and/or terminology edits have been made directly to the ms, via Word tracking feature; these edits must be made before the paper can be published.

The manuscript has been revised again by an English editing service and also contains your suggestions and comments, so that some sentences or paragraphs have been rewritten to be more concise. We also make some changes through the manuscripts as you can see at Lines 449-474 and Lines 477-487 suggested by reviewer 2.

  1. b) The authors need to be very careful regarding their use of generations, voltinism ecotypes; given their data, what they are seeing is a Mix of Both a Univoltine strain (that still appears to be the dominant strain), combined with a Multivoltine strain, that is known to have a facultative diapause (see the Palmer et al. 1985 ref.); this MV strain is most noticeable late season (Aug-Sept), and at this latitude of Romania, may only produce a partial moth flight, but it is consistent each year. (Much more could be said about this; but is beyond the scope of this study)

Key point: is be careful with the terminology; and be consistent throughout the paper

Thank you for your valuable comments and suggestions. It helped me to understand the existing situation in this part of Romania

With the revisions made to the attached Word file, the paper should be acceptable for publication.